# Alterations of Lipid Profile in Livers with Impaired Lipophagy

**DOI:** 10.3390/ijms231911863

**Published:** 2022-10-06

**Authors:** Wenke Jonas, Kristin Schwerbel, Lisa Zellner, Markus Jähnert, Pascal Gottmann, Annette Schürmann

**Affiliations:** 1Department of Experimental Diabetology, German Institute of Human Nutrition Potsdam-Rehbruecke, 14558 Nuthetal, Germany; 2German Center for Diabetes Research, 85764 München-Neuherberg, Germany; 3University of Potsdam, Institute of Nutritional Sciences, 14558 Nuthetal, Germany

**Keywords:** non-alcoholic fatty liver disease, lipophagy, lipidomics, fatty acid profile, long-chain polyunsaturated fatty acids

## Abstract

Non-alcoholic fatty liver disease (NAFLD) is characterized by excessive lipid accumulation in the liver. Various mechanisms such as an increased uptake in fatty acids or de novo synthesis contribute to the development of steatosis and progression to more severe stages. Furthermore, it has been shown that impaired lipophagy, the degradation of lipids by autophagic processes, contributes to NAFLD. Through an unbiased lipidome analysis of mouse livers in a genetic model of impaired lipophagy, we aimed to determine the resulting alterations in the lipidome. Observed changes overlap with those of the human disease. Overall, the entire lipid content and in particular the triacylglycerol concentration increased under conditions of impaired lipophagy. In addition, we detected a reduction in long-chain polyunsaturated fatty acids (PUFAs) and an increased ratio of n-6 PUFAs to n-3 PUFAs, which was due to the depletion of n-3 PUFAs. Although the abundance of major phospholipid classes was reduced, the ratio of phosphatidylcholines to phosphatidylethanolamines was not affected. In conclusion, this study demonstrates that impaired lipophagy contributes to the pathology of NAFLD and is associated with an altered lipid profile. However, the lipid pattern does not appear to be specific for lipophagic alterations, as it resembles mainly that described in relation to fatty liver disease.

## 1. Introduction

Non-alcoholic fatty liver disease (NAFLD) comprises a spectrum of disorders mainly characterized by the deposition of fat in hepatocytes without any impact in terms of alcohol consumption. The disease ranges from simple steatosis to non-alcoholic steatohepatitis, cirrhosis and hepatocellular cancer [1]. NAFLD is the most common chronic liver disease worldwide and is strongly associated with obesity, type 2 diabetes, metabolic syndrome and nutritional factors [2,3]. The prevalence of NAFLD is estimated to exceed 25% worldwide [4]. Not only adults, but also 3 to 10% of children have the characteristics of NAFLD [5], indicating that this disease spectrum is a major public health problem worldwide. 

The biological source of fatty acids which are taken up by the liver can primarily be linked to enhanced lipolytic activity in the adipose tissue (59%) [6]. In addition, 26% of triacylglycerols stored in the liver derive from increased hepatic *de novo* lipogenesis and only 15% from diet [6]. In addition to these mechanisms, an increased hepatic fat content can also be caused from impaired autophagy, notably lipophagy. Autophagy is a cellular self-digestive pathway that targets cytosolic components to lysosomes for degradation in order to maintain cellular homeostasis and supply substrates for energy generation [7,8]. So far, three different forms of autophagy have been described in mammalian cells: macroautophagy (referred to autophagy), microautophagy and chaperone-mediated autophagy [9]. As reported in 2009, autophagy was shown to mediate the degradation of intracellular lipid droplets, with this being termed lipophagy. Indeed, the inhibition of lipophagy is associated with an increased cellular triacylglycerol content, lipid droplet number and liver size [8].

A lipidomic analysis of liver biopsies from healthy people and NAFLD patients revealed differences in the abundance of specific lipid species [10,11,12], indicating that the composition of the lipid profile is changed upon NAFLD manifestation and progression. However, up until now, it is not known if and to which extent impaired lipophagy alters the hepatic lipid composition. We recently identified a quantitative trait locus (QTL) for fatty liver in the murine Collective Diabetes Cross [13] *Ltg/NZO* (liver triacylglycerols from NZO alleles). The locus contains a group of immunity-related GTPases (IRGs), two of which, *Ifgga2* and *Ifgga4*, were markedly suppressed in advanced NAFLD due to a single-nucleotide polymorphism in the FOXO2 binding site in the enhancer of both genes [14]. We recently demonstrated that IFGGA2 plays an important role in lipophagy. It is recruited from endosomes to lipid droplets when lipids accumulate in hepatocytes. There, IFGGA2 interacts with ATGL (adipocyte triglyceride lipase), which in turn binds the autophagy protein LC3B (microtubule associated protein 1 light chain 3 beta), leading to the induction of lipophagy and the prevention of excess hepatic lipid accumulation [14].

In the current study, we present a murine model of advanced NAFLD without confirmed inflammation, based on disturbed lipid catabolism due to impaired lipophagy. This led to changes in lipid metabolism that supported further fat storage in the liver and resulted in an altered lipid profile compared to the control group.

## 2. Results

### 2.1. Identified Locus Ltg/NZO on Chromosome 18 Increased Hepatic Fat Content

As previously described, the locus *Ltg/NZO* on chromosome 18 associates with increased hepatic triacylglycerols levels and is mediated by the suppression of the immunity-related GTPase *Ifgga2*, resulting in impaired lipophagy and increased hepatic fat accumulation [14]. To investigate differences in the lipid architecture of livers with an altered lipophagic capacity, an untargeted lipidome analysis was performed on livers of obese mice, which differ in their *Ifgga2* expression. The two recombinant congenic strains (RCSs), which were bred on the obese NZO background, differ genetically in terms of the 5.3 Mbp of the *Ltg/NZO* locus. Mice at the age of 7 weeks carrying one NZO and one C57BL/6 allele (*Ltg/NZO.5.3^N/B^*, referred to as IRG, immunity-related GTPases) and mice homozygous for the NZO allele (*Ltg/NZO.5.3^N/N^*, referred to as ΔIRG) were fasted for 16 h. To induce autophagy/lipophagy, a nutritional stressor such as prolonged fasting is required [7]. The IRG mice exhibited high *Ifgga2* expression while the ΔIRG mice displayed a very low *Ifgga2* expression in the liver (Figure 1A). Mice did not show any genotype-specific differences in body weight and liver weight at seven weeks of age (data not shown). However, the ΔIRG mice had significantly higher levels of hepatic triacylglycerols compared to the IRG mice (70.59 vs. 29.44 µg/mg tissue, Figure 1B).

### 2.2. Altered Hepatic Lipid Composition Induced by the Ltg/NZO Locus

Of the 941 lipid species measured by the differential mobility spectroscopy approach, 877 were detected in the current study. The measured lipid species can be divided into 14 lipid classes, which are classified into glycerolipids, phospholipids, sphingolipids and sterols (Table 1). The results are either presented as concentration (nmol/g), to provide quantitative information, or as composition (mol%), to investigate the relative quantity. As the liver of the ΔIRG mice accumulated significantly more lipids, we put a specific focus on the relative quantity in order to evaluate if impaired lipophagy affects the pattern of specific lipids. Lipidomics was performed on liver samples from 6- and 16-h-fasted mice. However, since we aimed at the effects of impaired lipophagy and the results of the samples after 6-h fasting showed only minor differences (data not shown), we focused our analysis on the 16-h fasted animals.

Between both groups, 29% of all measured lipid species were significantly different; 17% were more and 12% less abundant in the livers of the ΔIRG compared to those of the IRG mice (Figure 2A). Determining the lipid classes revealed significantly increased concentration of glycerolipids and sterols (cholesterol ester, CE) in livers with impaired lipophagy (ΔIRG) compared to controls (IRG), while the concentration of phospholipids and sphingolipids did not differ between mice (Figure 2B). Within the class of glycerolipids, the concentration of triacylglycerols (TGs) and diacylglycerols (DGs) in particular were significantly increased. The same effect was detected for CE. Among the sphingolipids, the levels of hexosylceramides (HexCers) were significantly lower in the livers of the ΔIRG mice in comparison to the IRG mice (Figure 2C). Glycerolipids such as DGs and TGs as well as CE are also most affected in human liver samples and increase with disease progression [15,16]. However, in contrast to our mouse data on HexCers, a recent liver lipidome analysis of more than 180 humans undergoing bariatric surgery detected increased levels of HexCers with increasing steatosis [15]. 

By comparing the relative abundance of the different major lipid classes, it is apparent that TG in particular were strongly increased in the ΔIRG mice relative to the other classes. This was accompanied by a relatively high decrease in phosphatidylcholines (PCs) and phosphatidylethanolamines (PEs) (Figure 2D).

### 2.3. Total Fatty Acid Composition Is Modulated by the Ltg/NZO Locus

The rough classification of fatty acids into saturated (SFAs), monounsaturated (MUFAs) and polyunsaturated (PUFAs) acids displayed an increased concentration in all three classes in the livers of the ΔIRG mice (Appendix A). However, in terms of composition, MUFAs accumulated in the ΔIRG mice in favor of SFAs and PUFAs (Figure 3A, left). Among the PUFAs, n-3 PUFAs were lower abundant in the ΔIRG mice (Figure 3A, middle) and the short-chain PUFAs (SC-PUFAs) had a higher percentage than the long-chain PUFAs (LC-PUFAs) in the ΔIRG mice (Figure 3A, right). Videla et al. suggested in a human study that the depletion of LC-PUFAs promotes fatty acid and TG synthesis rather than oxidation [17].

To determine whether the suppression of *Ifgga2* and impaired lipophagy affect the relative quantity of specific fatty acids, we analyzed their general distribution in total lipids as well as in the indicated classes (Figure 3B). The heat map shows that short-chain SFAs and n-3 PUFAs in TGs, DGs and PCs are lower in the ΔIRG mice, whereas PCs and PEs showed a reduction of the n-6 PUFAs C20:3 and C20:4. Most other n-6 PUFAs displayed a slight increase in ΔIRG livers (Figure 3B). This resulted in a significant increase of the n-6 to n-3 ratio and a decrease of the n-3 index (Figure 3C, upper panel), which have already been observed in NASH patients and are both markers of inflammatory processes during disease progression [18].

The most abundant saturated fatty acids in both genotypes within each lipid class were palmitic acid (C16:0) and oleic acid (C18:1; Appendix A). In TGs, C16:0 was significantly lower present and higher in DGs in ΔIRG livers (see below, Figure 3D, Appendix A). Palmitic acid originates from either the diet or lipogenesis and is the precursor of most n-7 and n-9 MUFAs synthesized through different elongation and desaturation steps (Figure 3D). By calculating the ratio of product to precursor, it is possible to estimate the activity of the fatty acid-synthesizing enzymes without having to determine them directly. Of course, this can only be considered as a surrogate marker, but it has proven to be reliable in other studies [11,19]. As indicated in Figure 3D, the estimated activity of stearoyl-CoA desaturase (SCD1, Δ9) determined from (C16:1/C16:0) and (C18:1/C18:0) was increased in the livers of the ΔIRG mice (Figure 3C, middle panel), which was accompanied by an accumulation of its products C16:1 and C18:1, respectively. However, further elongation to C24:1 or C24:0 as well as its catalyzing enzyme ELOVL fatty acid elongase 3 (ELOVL3) were decreased, resulting in an accumulation of C20:0 and C20:1 and a reduction in downstream products (Figure 3D and Appendix A).

Essential fatty acids such as alpha-linolenic (C18:3n-3, ALA) and linolenic (C18:2n-6, LA) acid are absorbed through food and are the precursors for other LC-PUFAs which cannot be synthesized by mammals. LC-PUFAs act as ligands for the peroxisome proliferator-activated receptor-α (PPARα), thereby upregulating genes for fatty acid oxidation. LC-PUFAs of the n-3 series are more effective inducers of PPARα expression than those of the n-6 series, although neither of them are strong inducers [20]. Although there were no differences in food intake between the genotypes (data not shown), both ALA and LA were enriched in the livers of the ΔIRG animals (Appendix A and Appendix A). LA was generally the predominant essential fatty acid. Due to similar or reduced fatty acid desaturase (FADS1, Δ5; FADS2, Δ6) and elongase (ELOVL5 and 6) activities (Figure 3C, lower panels), the overall amount of LC-PUFAs was reduced in the ΔIRG mice in comparison to the livers of the IRG mice (Figure 3A,E and Appendix A). In particular, arachidonic acid (C20:4n-6, AA) and docosahexanoic acid (C22:6n-3, DHA) were depleted in terms of total lipids (Figure 3B). Furthermore, applying the ratio proposed by Valenzuela et al. [21] to calculate the desaturase activity of the n-3 [C20:5n-3 (EPA)+DHA/ALA] and n-6 series (AA/LA) resulted in a 51% (*p* < 0.01) and 55% (*p* < 0.01) reduction, respectively (Figure 3).

We also assessed the expression of genes such as enzymes (e.g., *Elovl5* and *Scd1*) or central regulators (e.g., *Ppara*) involved in lipid metabolism. However, we did not measure any difference between strains after either a 6- or 16-h fast (Supplemental Appendix A).

While the indicated fatty acids were significantly regulated between both genotypes, the effect size was rather small (Appendix A). This can be attributed to the fact that the two obese mouse strains were genetically nearly identical except for a 5.3 Mbp region on chromosome 18 and that the study was performed at a young ageof the mice (7 weeks). Thus, it can be assumed that due to the reduced hepatic lipophagy caused by the *Ltg/NZO* locus, the ratio of SFAs, MUFAs and PUFAs was altered, whereby the depletion of LC-PUFAs in particular is worth noting. We believe that the influence of the locus could be even more pronounced at a later age, when the mice show an even more striking difference in liver fat content [14].

### 2.4. Triacylglycerols, the Lipid Class Mostly Affected by the Locus Ltg/NZO

The hallmark of fatty liver disease is the increased synthesis and accumulation of TGs within hepatocytes, which serve as energy storage and prevent the aggregation of toxic intermediates such as DG, thereby reducing lipotoxic stress. Indeed, most of the additional liver fat content in the ΔIRG mice can be attributed to elevated TG levels, whereas the overall DG levels were not altered in terms of composition (Table 1). This is also reflected in the TG/DG ratio, which was significantly increased in ΔIRG livers (IRG 32.6 ± 2.5; ΔIRG 53.2 ± 4.5; *p* < 0.01). 

Next, the impact of an altered lipophagic capacity on the PUFA-composition in TGs and DGs was analyzed. The most common fatty acids in TGs similar to the composition of the total fatty acid profile were palmitic acid (C16:0), which was decreased in ΔIRG livers, as well as oleic acid (C18:1) and linoleic acid (C18:2), which both increased in ΔIRG livers. The latter showed the same pattern in DGs, which, however, exhibited a higher degree of C16:0 in ΔIRG livers and thereby displayed the opposite effect to that of TGs (Figure 3B). The abundance of the sum of the different n3- and n-6 PUFAs was similar in TGs and DGs but ΔIRG mice showed significantly lower levels of n-3 PUFAs with no differences in the n-6 fatty acids. However, the ratio of n-6 to n-3 PUFAs was in both lipid classes significantly regulated, with higher levels in the ΔIRG mice (Figure 4A,B). Both eicosapentanoic acid (C20:5n-3, EPA) and DHA (C22:6n-3) were significantly depleted in TGs and DG, whereas AA (C20:4n-6) was only reduced in DGs (Figure 4C,D).

### 2.5. Changes in Phospholipid Classes

Phophatidylcholines (PC) and phosphatidylethanolamines (PEs) are the most abundant phospholipids of mammalian cells and cellular organelles and are known to be affected in obesity and NAFLD [22]. Interestingly, although the concentration of these lipid classes was only slightly different, *Ltg/NZO* appeared to affect the relative levels of PC and PE, as the ΔIRG mice had significantly lower percentages of these two phospholipid classes in their livers than the IRG mice (Table 1). The molar ratio of PC/PE, which is related to membrane integrity [23], where a decreased ratio promotes liver damage and may be involved in initiating inflammatory processes [24], was not different between the mice (IRG 1.17 ± 0.12; ΔIRG 1.15 ± 0.10). The overall analysis of total fatty acid composition revealed a reduction in n-3 PUFAs, including EPA and DHA (Figure 3B). The ΔIRG mice, which developed a more severe fatty liver, showed a significantly lower abundance of PEs containing EPA compared to mice with less severe steatosis (IRG). No genotype-specific difference could be detected for DHA in PCs or PEs (Figure 3B).

## 3. Discussion

The manifestation of NAFLD is associated with adverse metabolic changes, which include alterations in hepatic lipid composition [25]. It is not only the quantity of stored fat that has an impact on the disease, but also the quality [26]. In the current study, an unbiased lipidome approach was used to determine the specific lipid profile of mice characterized by increased hepatic lipid content due to impaired lipophagy [14].

The present mouse model for advanced NAFLD is not based on dietary differences but on genetics, caused by a fragment of 5.3 Mbp on chromosome 18 (IRG vs. ΔIRG) containing the cluster of immunity-related GTPases [14] which affects autophagy. Although there were only minor differences in lipid profiles between the IRG and ΔIRG mice, our analysis revealed several significant alterations. However, the results do not allow for the conclusion that the observed changes are specific to impairments in lipophagy. Most effects appear to be common in NAFLD because they correspond to those described in relation to the progression of fatty liver disease in humans: (i) an increase in DGs and TGs as well as CEs [16], resulting in an elevated TG/DG ratio [12], (ii) an overall reduction in the LC-PUFAs percentage [17] and an increased n-6/n-3 ratio driven by n-3 PUFA depletion [10,12,27,28], and (iii) a relative reduction in EPA (C20:5n-3) and DHA (C22:6n-3) in DGs and TGs [12] (Figure 5).

The finding of increased hepatic TG content with disease progression has already been confirmed in several studies and is considered a hallmark of NAFLD [10,12]. In general, four major pathways contribute to hepatic TG accumulation, including the elevated uptake of circulating fatty acids derived from the diet or adipose tissue lipolysis, increased hepatic fatty acid synthesis, the lower secretion of TGs via very low density lipoprotein (VLDL) particles and decreased fatty acid oxidation [29]. None of these pathways were affected by ΔIRG and the suppression of *Ifgga2*. Rather, an impaired lipophagy was discovered to be causative for increased hepatic TG storage in the ΔIRG mice [14]. Others have also described the impact of impaired lipophagy on NAFLD development in mice [30] and humans [31,32]. The increase in TGs, which are inert, can be seen as a defense mechanism against lipotoxicity induced by free fatty acids (FFAs), especially SFAs such as palmitic acid (C16:0) and stearic acid (C18:0). SFAs elicit lipotoxic effects through diverse mechanisms including the generation of reactive oxygen species [33], *de novo* ceramide synthesis and detrimental effects on mitochondrial function [34], which can eventually cause organ dysfunction and promote chronic inflammation. Overall, the progressive increase in SFAs correlates with disease severity in humans [11]. However, our analysis does not provide information on FFAs, but the concentration of SFAs in total lipids was increased in the ΔIRG mice, which implies the same for FFAs (see Appendix A).

DGs play a prominent role in NAFLD by acting as signaling molecules affecting insulin sensitivity [35]. The analysis of the fatty acid composition of DGs and TGs indicated higher percentages of palmitate (C16:0) and lower percentages of stearic acid (C18:0) in the DGs of livers with impaired lipophagy (ΔIRG), whereas in the TGs, the opposite effects were obtained. In a cardiomyoblast cell line, it was shown that palmitate induced DG accumulation mostly in the endoplasmatic reticulum (ER), which is associated with ER stress [36]. This may indicate that due to an impaired lipophagy-mediated lipid degradation, lipid turnover is affected, resulting in an enhanced package of palmitate into DGs. 

Another important observation of this study is the reduction in PCs and PEs in ΔIRG livers, an effect that is known to be associated with NAFLD progression [12] in mice and humans. Low levels of PEs and PCs are supposed to contribute to the disease by initiating inflammation or increasing the formation of large lipid droplets [22,23].

The reduction in LC-PUFAs, in particular of n3-PUFAs, in our model of impaired lipophagy is interesting. Earlier, it was shown that the treatment of hepatocytes with n-3 fatty acids such as EPA or DHA increased autophagic flux represented by an augmented LC3B-II/LC3B-I ratio and thereby preventing hepatocytes from lipotoxicity and increased lipid accumulation [37]. The authors showed that this effect was mediated by downregulating the expression of *Scd-1* in hepatocytes. Accordingly, ΔIRG mice with an impaired lipophagy exhibit an increased SCD1 activity, which was calculated by the (C16:1/C16:0) and (C18:1/C18:0) ratios. The desaturase SCD1 plays an essential role in lipid biosynthesis. It catalyzes the insertion of a cis double bond at the n-9 position into fatty acyl-CoA substrates including palmitoyl-CoA and stearoyl-CoA [38,39] and gives rise to a mixture of C16:1 and C18:1 unsaturated fatty acids [40]. In fact, both MUFAs are significantly upregulated in the lipids of the ΔIRG livers (Figure 3B).

### Limitations

Although we found significant differences, our study has certain limitations. The mice developed severe hepatic steatosis. However, they did not show the full spectrum of pathological features of NASH (non-alcoholic steatohepatitis), which include inflammation and fibrosis. This could be due to the relatively young age of the mice compared to other models of fatty liver disease [19,23]. Furthermore, since we did not include control mice with healthy livers, it is difficult to estimate how much the lipid profile deviates from the healthy state. In view of this, it is challenging to estimate the extent of changes due to impaired lipophagy.

## 4. Materials and Methods

### 4.1. Animals

Recombinant congenic mice (RCS) were generated as described [14]. RCS (*Ltg/NZO.5.3^N/B^*, referred to as IRG and *Ltg/NZO.5.3^N/N^*, referred to as ΔIRG) were phenotyped in the F4.N9 generation. Mice were kept under standard conditions (22 °C, 12-h light/12-h dark photoperiod), with the light switched on at 6:00 am (Zeitgeber time 0, ZT0 corresponding to the environmental circadian time set). At three weeks of age, animals received a high-fat diet (HFD) (D12451, Research Diets Inc., New Brunswick, NJ, USA). At the age of seven weeks, mice were sacrificed after a 16-h fast from ZT12 to ZT28. Plasma and organs were frozen in liquid nitrogen and stored at −80 °C for biochemical analyses.

Animal experiments were performed referring to the ARRIVE guidelines and approved by the ethics committee of the State Agency of Environment, Health, and Consumer Protection (State of Brandenburg, Germany 2347-10-2014).

### 4.2. Gene Expression Analysis

The total RNA from livers was isolated using RNeasy Mini Kits (Qiagen, Hilden, Germany) and cDNA prepared by a M-MLV Reverse Transcriptase-Kit (Promega, Madison, WI, USA). Genome-wide expression analysis from mice after a 6-h fast was performed by OakLabs GmbH (Germany, Hennigsdorf, Germany) applying SurePrint G3 Mouse GE 8x60k Microarray gene chips (Agilent Technologies, Santa Clara, CA, USA), as previously described [14]. Expression analysis for livers of 16 h-fasted mice was performed by applying hydrolysis probes (Supplemental Appendix A) for quantitative reverse transcription PCR (qRT-PCR). *Eef2*, *Ppia* and *Hprt* were used as reference genes. Relative gene expression was analyzed using the ΔCt method [41].

### 4.3. Detection of Liver Triacylglycerides 

Liver triacylglycerols were quantified enzymatically using a commercial kit (RandoxTR-210, Crumlin, United Kingdom), as described previously [13]. In brief, livers were homogenized for 5 min in 10 mmol/L sodium dihydrogen phosphate, 1 mmol/L EDTA and 1% (*v*/*v*) polyoxyethylene-10-tridecyl ether and incubated for 5 min at 70 °C to inactivate enzymes. After an additional 5 min incubation on ice, the samples were centrifuged and the supernatant was diluted 1:10 due to turbidity and a fat layer on top. The diluted samples were incubated at 70 °C and on ice for 5 min each. After the second centrifugation, the clear supernatant was used for the enzymatic assay following the manufacturer’s protocol.

### 4.4. Lipidomic Analysis

Lipidomics analysis of hepatic tissue was performed by Metabolon (Morrisville, NC, USA) using the Metabolon TrueMass^®^ Complex Lipid Panel. Samples were prepared as follows: lipid fraction was extracted with dichloromethane:methanol overnight at 4 °C. Supernatants were subjected to a modified Bligh–Dyer extraction using methanol/water/dichloromethane in the presence of deuterated internal standards. For data acquisition, sample extracts were dried under nitrogen and reconstituted in a dichloromethane:methanol solution containing ammonium acetate. Extracts were transferred to vials for infusion-MS analysis, performed on a Shimadzu LC with nano PEEK tubing and the Sciex SelexIon^®^-5500 QTRAP. The samples were analyzed via both positive and negative mode electrospray. The 5500 QTRAP was operated in multi reaction monitoring (MRM) mode with a total of more than 1100 MRMs. Individual lipid species were quantified by taking the ratio of the signal intensity of each target compound to that of its assigned internal standard, then multiplying this by the concentration of the internal standard added to the sample. Lipid class concentrations were calculated from the sum of all molecular species within a class, and fatty acid compositions (mol%) were determined by calculating the proportion of each class comprised by individual fatty acids. The lipidomic analysis does not provide any information on free fatty acids. For triacylglycerols, only one esterified fatty acid was specified.

In total, 941 different lipid species were analyzed belonging to 14 different lipid classes (cholesteryl esters, CEs; monoacylglycerols, MGs; ceramides, Cers; dihydroceramides, dhCers; lactosylceramides, LacCers; hexosylceramides, HexCers; sphingomeylins, SMs, lysophosphatidylethanolamines, LPEs; lysophosphatidylcholines, LPCs; diacylglycerols, DGs; triacylglycerols, TGs; phosphatidylcholines, PCs; phosphatidylethanolamines, PEs; and phosphatidylinositol, PI).

### 4.5. Statistical Analysis

Statistical analysis of two groups was performed by unpaired *t*-test with Welch’s correction. Data are presented as means ± SD. *p* ≤ 0.05 was regarded as statistically significant, and the results were calculated with Prism 8 (GraphPad Software, La Jolla, CA, USA).

## Figures and Tables

**Figure 1 ijms-23-11863-f001:**
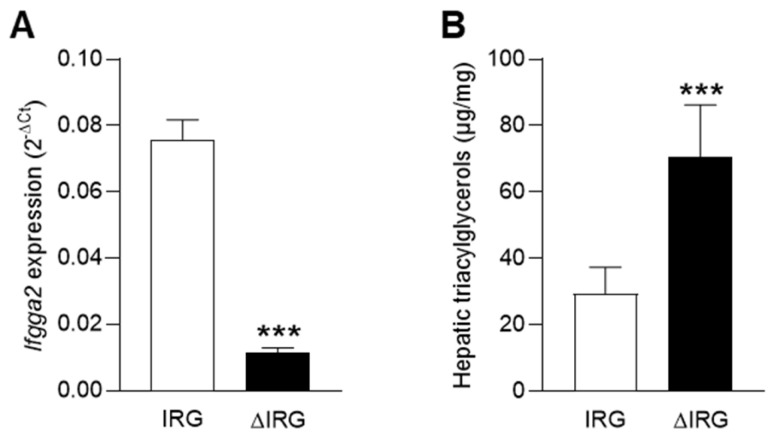
Mice with impaired lipophagy store more fat in the liver. (**A**) Expression of the lipophagy-associated *Ifgga2* in livers of mice that were heterozygous (IRG) or homozygous (ΔIRG) for the locus *Ltg/NZO*. (**B**) Hepatic triacylglycerol levels of IRG and ΔIRG mice. Data are shown as means ± SD and analyzed by unpaired *t*-test with Welch’s correction. *** *p* < 0.001; IRG, *n* = 9; ΔIRG, *n* = 7.

**Figure 2 ijms-23-11863-f002:**
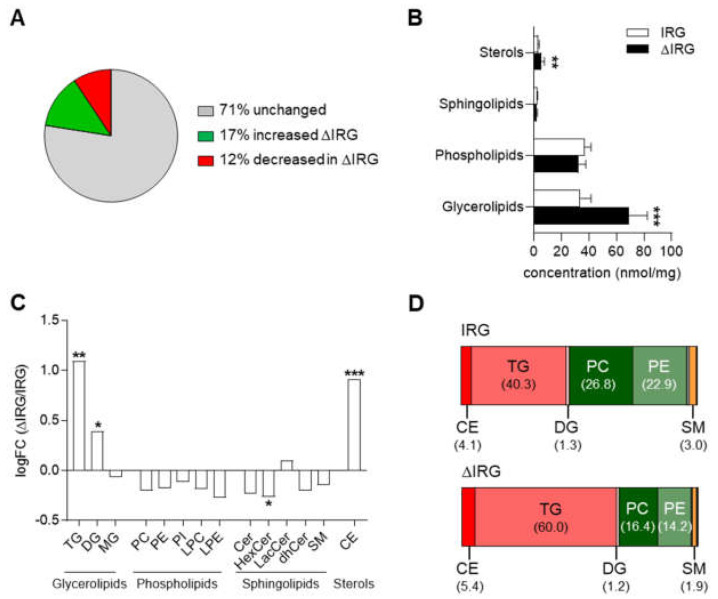
Mice with impaired lipophagy accumulate more glycerolipids and sterols in the liver. (**A**) Percentage of significantly regulated lipid species in ΔIRG and IRG mice. (**B**) Concentration of glycerolipids, phospholipids, sphingolipids and sterols in IRG and ΔIRG mice. (**C**) Concentration of lipid classes in ΔIRG vs. IRG mice shown as logFC. (**D**) Relative abundance of major lipid classes in IRG (top) and ΔIRG (bottom) mice. Data are shown as means ± SD and analyzed by unpaired *t*-test with Welch’s correction or one sample *t*-test vs. 0. * *p* < 0.05; ** *p* < 0.01; *** *p* < 0.001. IRG, *n* = 7; ΔIRG, *n* = 7.

**Figure 3 ijms-23-11863-f003:**
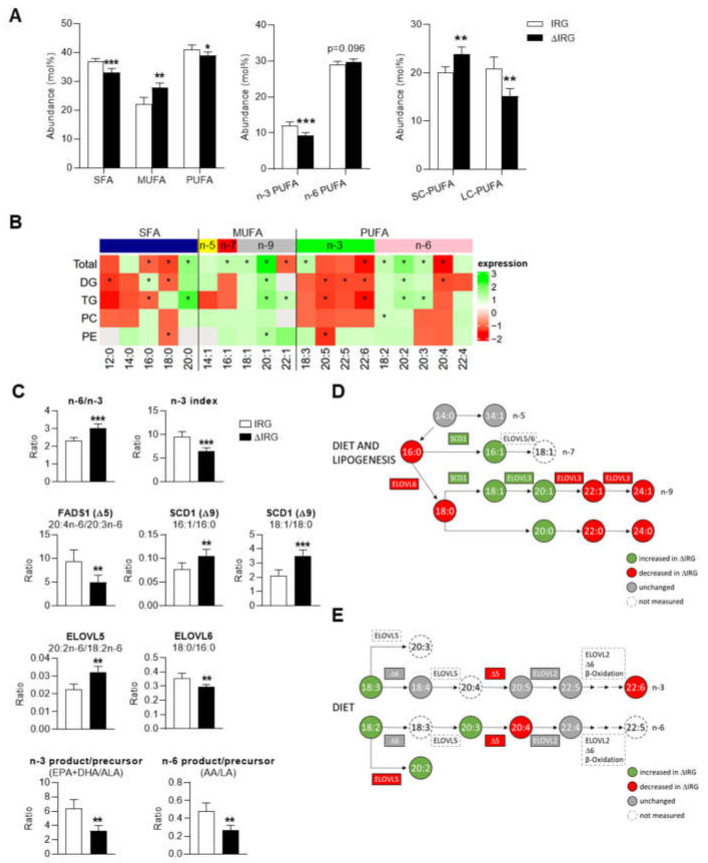
Impact of altered lipophagic capacity on fatty acid composition. (**A**) Relative quantity of saturated (SFAs), monounsaturated (MUFAs) and ployunsaturated (PUFAs) fatty acids levels (**left**). Abundance of n-3 and n-6 PUFAs (**middle**) and analysis of PUFAs in regard of chain length (**right**). (**B**) Heat map presenting the distribution of different fatty acid species within total fatty acids (top row) and individual lipid classes in ΔIRG and compared to IRG mice. * indicates significant differences. Not detected fatty acids are marked in grey. (**C**) Evaluation of enzyme activities based on product to precursor ratios. The n-6/n-3 ratio includes the fraction of all fatty acids that belong to either the n-3 or the n-6 series. The n-3 index comprises all n-3 fatty acids. Schemes for the biosynthesis of (**D**) SFAs and MUFAs as well as (**E**) PUFAs. Visualization adapted from mapping tool Lipid Surveyor™ by Metabolon. Data are shown as means ± SD and analyzed by unpaired *t*-test with Welch’s correction. * *p* < 0.05; ** *p* < 0.01; *** *p* < 0.001; IRG, *n* = 7; ΔIRG, *n* = 7.

**Figure 4 ijms-23-11863-f004:**
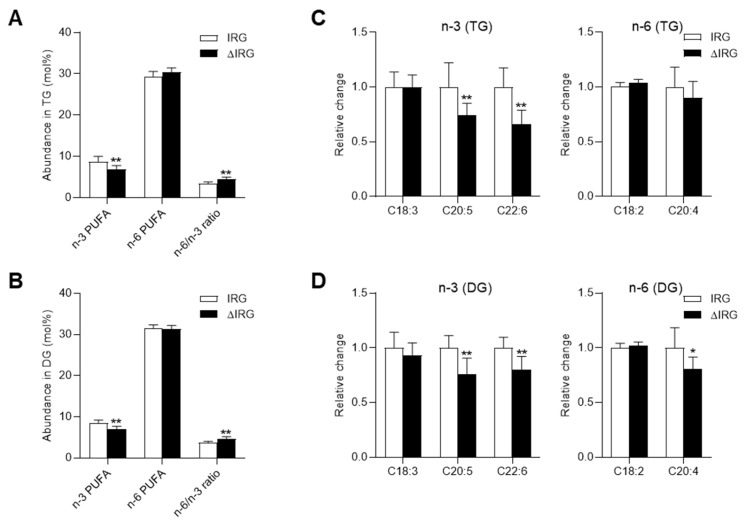
Polyunsaturated fatty acid composition in di- and triacylglycerols. Overall proportion of n-3 and n-6 PUFAs in the (**A**) triacylglycerols (TG) and (**B**) diacylgylcerols (DG) lipid classes in ΔIRG mice in comparison to IRG mice. Relative changes of specific n-3 and n-6 PUFAs in the (**C**) TG and (**D**) DG lipid classes of ΔIRG mice. Data are shown as means ± SD and analyzed by unpaired *t*-test with Welch’s correction. * *p* < 0.05; ** *p* < 0.01; IRG, *n* = 7; ΔIRG, *n* = 7.

**Figure 5 ijms-23-11863-f005:**
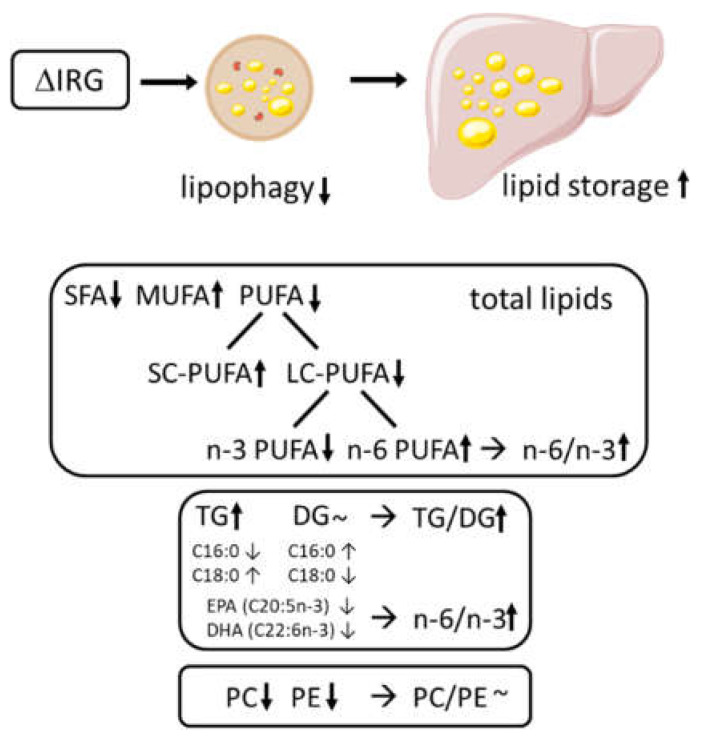
Decreased lipophagy in ΔIRG mice resulted in increased lipid storage in the liver and changes in the hepatic lipidome. The main effects included an overall reduction in LC-PUFA content and an increased n-6/n-3 ratio, which was due to the depletion of n-3 PUFAs and an increased TG/DG ratio. Although the phospholipid classes PC and PE were reduced, the PC/PE ratio was not affected.

**Table 1 ijms-23-11863-t001:** Hepatic lipid content in mice with reduced lipophagy (ΔIRG) compared to control mice (IRG) after a 16-h fast. Absolute (concentration, nmol/mg) and relative (composition, mol%) values for individual lipid classes are presented. Data are shown as means ± SD and analyzed by unpaired *t*-test with Welch’s correction. Significant values (*p* < 0.05) are bold. IRG, *n* = 7; ΔIRG, *n* = 7.

Lipid Class	Abbr.	Species in Class	Concentration (nmol/mg)		Species		Composition (mol%)		Species	
			IRG	ΔIRG	Down	Up	IRG	ΔIRG	Down	Up
**Glycerolipids**										
Triacylglycerol	TG	518	29.687 ± 7.193	**63.494 ± 12.995**	0	370	40.292 ± 6.894	**59.986 ± 5.720**	82	109
Diacylglycerol	DG	58	0.921 ± 0.162	**1.212 ± 0.201**	1	17	1.256 ± 0.169	1.165 ± 0.209	13	4
Monoacylglycerol	MG	22	0.162 ± 0.062	0.155 ± 0.076	0	1	0.228 ± 0.108	0.150 ± 0.071	0	0
**Phospholipids**										
Phosphatidylcholine	PC	72	19.502 ± 2.593	16.940 ± 2.959	7	0	26.799 ± 3.913	**16.381 ± 3.477**	0	5
Phosphatidylethanolamine	PE	93	16.657 ± 2.093	14.721 ± 2.332	9	4	22.857 ± 2.975	**14.161 ± 2.388**	3	9
Phosphatidylinositol	PI	7	0.252 ± 0.042	0.233 ± 0.042	0	0	0.348 ± 0.069	**0.222 ± 0.029**	1	1
Lysophosphatidylcholine	LPC	16	0.572 ± 0.136	0.502 ± 0.090	1	0	0.788 ± 0.195	**0.483 ± 0.088**	0	0
Lysophosphatidyl- ethanolamine	LPE	11	0.071 ± 0.019	0.059 ± 0.017	0	0	0.097 ± 0.027	**0.056 ± 0.013**	0	0
**Sphingolipids**										
Ceramide	Cer	12	0.126 ± 0.027	0.107 ± 0.021	1	0	0.173 ± 0.039	**0.102 ± 0.014**	1	2
Hexosylceramide	HexCer	10	0.025 ± 0.002	**0.021 ± 0.003**	3	2	0.034 ± 0.004	**0.020 ± 0.005**	1	4
Lactosylceramide	LacCer	10	0.003 ± 0.000	0.003 ± 0.000	0	1	0.004 ± 0.001	**0.003 ± 0.001**	2	1
Dihydroceramide	dhCer	10	0.022 ± 0.002	0.019 ± 0.003	2	1	0.030 ± 0.004	**0.018 ± 0.002**	0	2
Sphingomyelin	SM	12	2.178 ± 0.281	1.964 ± 0.311	1	0	2.992 ± 0.429	**1.887 ± 0.293**	0	2
**Sterols**										
Cholesteryl ester	CE	26	3.018 ± 0.847	**5.678 ± 1.892**	0	23	4.102 ± 1.033	5.365 ± 1.461	5	11

## Data Availability

All data used in this manuscript are available upon request.

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
