# Peer review of "Alterations of Lipid Profile in Livers with Impaired Lipophagy"

_ijms, 2022, doi:10.3390/ijms231911863_

Round 1

Reviewer 1 Report

The authors provide a lipidomics study of the differences caused by the lower expression of Ifgga2.

 The study is a sufficient complement to the 2020’s study in Journal of Hepatology.

Why the mice have been fasted for 16h? Since signaling changes dramatically due to those fasting periods. What was the biological need, since it is not a physiological situation to have those long fasting periods, neither in rodents nor in humans? What was the period of fasting (ZT)? When the mice were sacrificed?

The calculations of enzyme activities (Fig. 3) could be supported by immunohistochemical staining of central enzymes (FAS, SCD1, ELOVL, ...). Also e.g. qPCR measurements of central transcriptions factors and the levels of underlying hormones and regulators, e.g. insulin, ATGL, HSL would provide insights in the mechanisms of the different regulations.

Line 57, 60, 61: Explanation for abbreviation is missing

Author Response

Reviewer #1

Comments and Suggestions for Authors

The authors provide a lipidomics study of the differences caused by the lower expression of Ifgga2.

 The study is a sufficient complement to the 2020’s study in Journal of Hepatology.

Why the mice have been fasted for 16h? Since signaling changes dramatically due to those fasting periods. What was the biological need, since it is not a physiological situation to have those long fasting periods, neither in rodents nor in humans? What was the period of fasting (ZT)? When the mice were sacrificed?

The mice were fasted for 16 h, as our aim was to induce lipophagy and to study to what extent the hepatic lipid profile differs between mice with normal and those with low expression of Ifgga2, a novel regulator of lipophagy. For the induction of autophagy/lipophagy, a prolonged fasting is required (Bagherniya et al., 2018) and this is for instance reached by intermittent fasting which is nowadays performed by a lot of people. However, we also performed an untargeted lipidome analysis of the two mouse strains after a 6 h fasting period. The general effects such as the higher abundance of lipid classes like tri- and diacylglycerols as well as ceramides were already observed after 6 hours of fasting. However, the impact was less pronounced. In the revised manuscript we provide an explanation for the long fasting period and state that no significant differences of the lipidome were detected after a 6 h fasting period. The 16-hour fasting period began with the onset of the dark phase (6 pm). Therefore, the mice were sacrificed around 10 am.

The calculations of enzyme activities (Fig. 3) could be supported by immunohistochemical staining of central enzymes (FAS, SCD1, ELOVL, ...). Also e.g. qPCR measurements of central transcriptions factors and the levels of underlying hormones and regulators, e.g. insulin, ATGL, HSL would provide insights in the mechanisms of the different regulations.

To approach this question, we measured the expression of several genes involved in fatty acid metabolism by qRT-PCR in liver samples from the 16 h fasted mice and compared the expression data with a transcriptomic analysis of liver samples from 6 h fasted mice from our original study (Schwerbel et al., 2020). We did not detect any differences between our mouse strains after a 6- or 16-hour fast which is something we expected because most of the proteins involved in the regulation of lipid metabolism are regulated on the level of activity and not on the level of expression. We have included the dataset in the supplementary file as Figure S3.

In our opinion, it does not make sense to measure expression of transcription factors and levels hormones because our aim was not to study how lipophagy is activated but how the lipidome differs in the liver when lipophagy is impaired as it is the case in ∆IRG mice.

Line 57, 60, 61: Explanation for abbreviation is missing

We included the explanations for the abbreviations SNP = single nucleotide polymorphism; ATGL = adipocyte triglyceride lipase and LC3B = microtubule associated protein 1 light chain 3 beta

Reviewer 2 Report

Authors evaluated the changes of lipid profiles in impaired lipophagy mice model. Present data were interesting and useful for clinicians to consider the pathogenesis of NAFLD. In present work, lipid profiles alteration was clearly shown. But lack of liver pathology, autophagy in liver and fat degeneration of hepatocytes diminish the significance. This murine model is quite different from human NASH. Authors should discuss it as limitations.     

Author Response

Reviewer 2

Authors evaluated the changes of lipid profiles in impaired lipophagy mice model. Present data were interesting and useful for clinicians to consider the pathogenesis of NAFLD. In present work, lipid profiles alteration was clearly shown. But lack of liver pathology, autophagy in liver and fat degeneration of hepatocytes diminish the significance. This murine model is quite different from human NASH. Authors should discuss it as limitations.     

We would like to thank reviewer #2 for the positive feedback. Concerning the criticism that the study lacks liver pathology we have to point out that we described the molecular function of Ifgga2, its interaction with the lipase ATGL and its role in the induction of lipophagy in our first publication in the Journal of Hepatology in 2020 (Schwerbel et al., 2020). We referred to the original findings in the introduction and the discussion. However, we agree that the ∆IRG mouse is different from human NASH. In our model, the mice develop severe hepatic steatosis over time, which is detectable already at a very early age. However, the established fatty liver does not progress further to inflammation or even fibrosis. This could be due to the relatively young age of the mice. In dietary mouse models for the disease, the animals are fed a high-fat diet for a much longer period of time. However, this is not possible the IRG and ∆IRG mice because they are on the obese NZO background. NZO mice develop severe hyperglycaemia and would die from a type 2 diabetes-like phenotype before the full pathology of fatty liver disease could develop.

We have added a paragraph mentioning the limitations of this mouse model at the end of the discussion.

Round 2

Reviewer 1 Report

Thanks for the comprehensive answers and the supplementation of the data!

Please state the period of fasting and sacrifice in zeitgebener time (ZT) in the methods section.

Author Response

Dear reviewer 1,

We would like to thank you very much for evaluating our revised manuscript (ijms-1921868) entitled:

“Alterations of lipid profile in livers with impaired lipophagy”.

Comments and Suggestions for Authors 

Thanks for the comprehensive answers and the supplementation of the data!

Please state the period of fasting and sacrifice in zeitgebener time (ZT) in the methods section.

We have now included in the methods section the fasting period and the sacrifice time according to Zeitgeber time.

Reviewer 2 Report

Revised manuscript was well addressed to reviewer's comments, and well written. I think it is acceptable for publication.

Author Response

Dear Reviewer,

thank you for the positive feedback on our revised manuscript.